



# Effects of atmospheric circulations on the interannual variation in PM$_{2.5}$ concentrations over the Beijing-Tianjin-Hebei region in 2013-2018

Xiaoyan Wang*[1,2], Renhe Zhang[1,2]

1. Department of Atmospheric and Oceanic Sciences & Institute of Atmospheric Sciences, Fudan University, Shanghai, China

2. Big Data Institute for Carbon Emission and Environmental Pollution, Fudan University, Shanghai, China

Correspondence to: wangxyfd@fudan.edu.cn

## Abstract

The Chinese government has made many efforts to mitigate fine particulate matter (PM$_{2.5}$) pollution in recent years by taking strict measures on air pollutants reduction, which has generated the nationwide improvements in air quality since 2013. However, under the stringent air pollution controls, how PM$_{2.5}$ concentration varies and how much the meteorological conditions contribute to the interannual variations in PM$_{2.5}$ concentrations are still unclear, which is very important for the local government to assess the emission reduction of previous year and adjust mitigation strategies of next year. The effects of atmospheric circulation on the interannual variation in wintertime PM$_{2.5}$ concentrations over the Beijing-Tianjin-Hebei (BTH) region in the period of 2013-

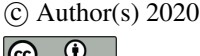



2018 are evaluated in this study. Generally, the transport of clean and dry air masses and unstable
boundary layer working with the effective near-surface horizontal divergence or pumping action at
the top of the boundary layer benefit for the horizontal or vertical diffusion of surface air pollutants.
Instead, the co-occurrence of a stable boundary layer, frequent air stagnation, positive water vapor
advection and deep near-surface horizontal convergence exacerbate the air pollution. Favorable
circulation conditions lasting for 2~4 days are beneficial for the diffusion of air pollutants, and 3~7
days of unfavorable circulation events exacerbate the accumulation of air pollutants. The occurrence
frequency of favorable circulation events is consistent with the interannual variation in seasonal
mean $PM_{2.5}$ concentrations. There is better diffusion ability in the winters of 2014 and 2017 than in
other years. A 76.5% of the observed decrease in $PM_{2.5}$ concentrations in 2017 over the BTH region
could be attributed to the improvement in atmospheric diffusion conditions. It is essential to exclude
the contribution of meteorological conditions to the variation in interannual air pollutants when
making a quantitative evaluation of emission reduction measurements.

## Introduction

Rapid economic development and associated emissions have led to recent severe air pollution over
China, which has become a central issue of concern for the public and governments (Wang et al.,
2018; Zhang et al., 2014; Song et al., 2018; Mu and Zhang, 2014; Tao et al., 2018). High levels of
fine particulate matter ($PM_{2.5}$) concentrations influence people's daily lives and threaten public
health (Liu et al., 2019; Zhao et al., 2018a; Hong et al., 2019; Zhang et al., 2017). In addition, they
are efficient in scattering and absorbing solar radiation, and are involved in the climate change by
changing the surface energy budget (Wang et al., 2009; Wang et al., 2017; Bi et al., 2016; Chen et al.,
2019b; Li et al., 2018; Zhao et al., 2019b; Che et al., 2019). To mitigate $PM_{2.5}$ pollutions, the Chinese
government issued the Air Pollution Prevention and Control Action Plan (hereinafter referred to as
the Clean Air Action hereinafter) in 2013, which required the Beijing-Tianjin-Hebei (BTH) region,
Yangtze River Delta and Pearl River Delta to reduce their $PM_{2.5}$ concentrations by 15~25% from
2013 to 2017 (China's State Council, 2013). A series of stringent clean air actions was implemented
to improve air quality, including strengthening industrial emission standards, phasing out small and



polluting factories, strengthening vehicle emission standards and more (Zhang and Geng, 2019). To
further improve air quality, the state council has released a three-year action to win the battle for a
blue sky in 2018, solidifying a timetable and roadmap for improving air quality. By 2020, emissions
of sulfur dioxide and nitrogen oxides are required to decline by at least 15% from 2015 levels, while
cities with low air quality standards should see their $PM_{2.5}$ density fall by at least 18%, according to
the plan (China's State Council, 2018). To achieve these goals, many efforts have focused on
adjustments to industrial, energy and transportation structures involved with central to local
government.
With the implementation of the toughest-ever clean air actions from Clean Air Action, the
anthropogenic emissions show significant decreased by 59% for $SO_2$, 21% for NOx, 23% for CO,
36% for $PM_{10}$ and 33% for primary $PM_{2.5}$ from 2013 to 2017 (Zheng et al., 2018; Wang et al., 2019b).
As a consequence, air quality in China improved significantly in terms of annual mean $PM_{2.5}$
concentrations, polluted days and pollution durations from 2013 to 2017, and surpassed the
mitigation targets of the Clean Air Action (Fan et al., 2020; Zhao et al., 2018b; Gui et al., 2019;
Zhong et al., 2018). By the end of 2017, the BTH region achieved its primary goal of reducing the
annual average $PM_{2.5}$ concentration to less than 60 $\mu g/m^3$ with a decreasing trend of $-9.3 \pm 1.8$ $\mu g/m^3$
(Wang et al., 2019b). However, in addition to air pollutants emissions, atmospheric meteorological
conditions play an important role in the transport, accumulation, scavenging and chemical
production of particles, which drives the evolution of every air pollution episode (Zhang et al., 2012;
Leung et al., 2018; Huang et al., 2018; Wang et al., 2016). Moreover, the interannual to interdecadal
variations in meteorological or climate signals (e.g., monsoon intensity, variation in sea ice, and the
occurrence of El Niño Southern Oscillation (ENSO) and North Atlantic Oscillation (NAO)) also
have significant effects on the variation in ambient $PM_{2.5}$ concentrations (Li et al., 2016; Chen et
al., 2019a; Zhao et al., 2018c; Dang and Liao, 2019; Feng et al., 2019; Yin et al., 2019). The global
warming associated with climate change may also contribute to the air pollution in China (Cai et al.,
2017; Zhang, 2017).
Recently, many researchers investigated how much of the recent decreased $PM_{2.5}$ concentrations
could be attributed to the contribution to emission reduction compared to the effects of atmospheric



elements. The studies have been carried out to evaluate the relative effects of emission reduction
and meteorological conditions on the recent decrease in $PM_{2.5}$ concentrations (Ding et al., 2019;
Guo et al., 2019; He et al., 2018; Zhang et al., 2019c; Zhao et al., 2019a). Based on a multiple linear
regression model, 12% of the decreased $PM_{2.5}$ over China is due to favorable meteorological
conditions between 2013 and 2018 (Zhai et al., 2019).  For the BTH region, Zhang et al. (2019b)
used the parameter linking air quality and meteorology (PLAM) index (a meteorological pollution
index for air quality) to evaluate meteorological conditions, and found that only approximately 5%
of the 39.6% reduction in $PM_{2.5}$ in 2017 could be attributed to meteorological changes. The relative
contribution of emission reduction to the decreased $PM_{2.5}$ concentrations in Beijing calculated by
the statistical model and Weather Research and Forecasting-Community Multiscale Air Quality
(WRF-CMAQ) was 80%, indicating that emission reductions were crucial for air quality
improvement in Beijing from 2013 to 2017 (Chen et al., 2019c). In addition, Zhang et al. (2019a)
quantified the contribution of different emission control policies to the rapid improvement in $PM_{2.5}$
pollution over China from 2013 to 2017 and highlighted the significant effects of strengthening
industrial emission standards and upgrading industrial boilers on air quality improvement during
the Clean Air Action.
Based on the investigation of different methods, the effectiveness of emission mitigation actions
was confirmed to drive the recent remarkable improvement in air quality in China since 2013.
However, most of the existing studies have focused on the relative long-term variation of air quality
(i.e., five to six years since 2013) and evaluated emission reduction effects over a multiyear time
scale. The Chinese government took a series of steps to reduce air pollutant emissions, which
requires a certain sacrifice regarding economic growth. In this situation, the local government need
an accurate evaluation of the emission reduction effects during the previous year and reasonable
adjustment of the mitigation policies of next year to keep the balance of economic growth and
environmental protection. The accurate evaluation of emission reduction effects should exclude the
meteorological element contribution to the interannual variations of air quality. Therefore, the
contribution of meteorological conditions to the interannual variation in wintertime $PM_{2.5}$
concentrations over the BTH region will be discussed in this study.






## 2. Data and Methods

### 2.1 On-site PM$_{2.5}$ mass concentration

The wintertime (December to February of the following year) hourly observed PM$_{2.5}$ mass concentration dataset over China from 2013 to 2018 was provided by the Ministry of Ecology and Environment of the People's Republic of China (http://106.37.208.233:20035). This study mainly focuses on the region of BTH region (113.5º-119ºE and 36º-42.5ºN, the solid-line box in Fig. 3), and 114 PM$_{2.5}$ stations are available over this region. The hourly PM$_{2.5}$ concentration was averaged to the daily mean value with no more than 40% missing data.

### 2.2 Method of atmospheric circulation classification

Commonly used objective classification methods include correlation, clustering, nonlinear methods, principal component analysis (PCA), and fuzzy analysis. Huth et al. (2008) compared these five classification methods and proposed that the performance of the T-mode PCA was the best in terms of its reproduction of predefined types, temporal and spatial stabilities, and reduced dependence on preset parameters. The T-mode PCA has been successfully applied to studies of general circulation models (Huth, 2000), climate change (Cavazos, 2000), and local air pollution (Xu et al., 2016; Valverde et al., 2015; Miao et al., 2017; Li et al., 2019). Zhang et al. (Zhang et al., 2012) first employed the obliquely rotated T-mode PCA method developed by COST action 733 (http://www.cost733.org) (Philipp et al., 2014) to identify the circulation pattern that is conductive to PM pollution in North China. In this study, the four-times-daily dataset of the fifth generation European Centre for Medium-Range Weather Forecasts (ECMWF ERA5) atmospheric reanalysis in winters from 2013 to 2018 with a horizontal resolution of 0.25º was used for synoptic circulation classification. The daily mean geopotential height fields at 925, 850 and 500 hPa were applied to the T-mode PCA method in the Cost733 toolbox. Our target region is 105º-125ºE and 30º-55ºN (the dashed box in Fig. 3).





**2.3 Model simulation**
The regional chemical/transport model WRF chemical model (WRF-Chem) version 4.0, was
applied to simulate the effects of meteorological condition variation on seasonal air pollution over
northern China at a horizontal resolution of 9 km (245*220 horizontal grid cells) and vertical
resolution of 33 layers. The simulation domain covers most areas of the North China region (Fig.
10). The initial and lateral meteorological boundary conditions are derived from the NCEP FNL
reanalysis data every 6 hours. The chemical and aerosol mechanisms used were the RADM2
chemical mechanism from Stockwell et al. (1990) and MADE/SORGAM aerosols (Ackermann et
al., 1998; Schell et al., 2001). MADE/SORGAM are used to simulate all major aerosol components
including sulfate, nitrate, ammonium, black carbon, organic carbon, sodium, chloride, mineral dust,
and water content. Madronich photolysis was used to calculate photochemical reactions. Other
major physical processes included the CAM shortwave radiation (Collins et al., 2004), RRTMG
longwave radiation (Iacono et al., 2008), the unified Noah land-surface model land surface option
and MYJ planetary boundary layer parameterization (Janjić, 1994).
To evaluate the impacts of meteorological contributions on the $PM_{2.5}$ variation between the 2016
winter (Dec. 2016 to Feb. 2017) and 2017 winter (Dec. 2017 to Feb. 2018) over the BTH region,
we conducted two sensitivity runs: the same emissions as the 2016 winter and the actual
meteorological conditions of 2016 and 2017. Thus, the difference in the simulated $PM_{2.5}$
concentrations between the 2016 and 2017 winters could be attributed to the meteorological
variation. The anthropogenic emission inventory for 2016 developed by Tsinghua University was
used in this study (available at http//www.meicmodel.org), as is named the Multiresolution
Emission Inventory for China (MEIC), containing monthly anthropogenic emissions of $SO_2$, NOx,
CO, NH3, $PM_{2.5}$, PMcoarse, BC, OC and NMVOCs. The horizontal resolution of the MEIC used
in this study is 0.25°. Each simulation is initialized at 00:00 UTC on Nov. 15, and the first 15-day
simulations are regarded as the spin-up period. Daily mean $PM_{2.5}$ concentrations between Dec. 1,
2016 to Feb. 28, 2017, and Dec. 1, 2017 to Feb. 28, 2018, are used to investigate the effects of
meteorological conditions on seasonal air pollution.



## 3. Results

### 3.1 Dominate synoptic circulation types in winter over the BTH region

As shown in Fig.1, the wintertime $PM_{2.5}$ concentrations over the BTH region show a remarkable decrease from 2013 to 2018 due to a series of air pollution reduction measures. Compared to 2013, the mean $PM_{2.5}$ concentration for 2018 decreased by 35.6% over 114 stations around the BTH region (cf. Table 1). However, under the background of improved air quality, evident interannual variations in $PM_{2.5}$ concentrations have been observed in recent years. The $PM_{2.5}$ concentrations in the winters of 2016 and 2018 are higher than those in the same period of the previous year, with mean values increasing by 18% and 13.36%, respectively. The high emissions of primary fine particulate matters and its precursors are considered as internal factors of severe $PM_{2.5}$ pollution in China; thus, emission reduction is the most direct and effective way to improve local air quality. However, the evolution of each air pollution episode is strongly affected by the local synoptic circulation pattern. Both emissions and atmospheric conditions are related to the ambient $PM_{2.5}$ concentration level. It is essential to exclude the atmospheric circulation impacts on air quality when assessing emission mitigation effects.

We use synoptic circulation types to measure the ability of atmospheric circulation to the accumulate, remove, and transport air pollutants. The daily mean geopotential height fields at 925, 800 and 500 hPa in the winters of 2013 to 2018 (total of 451 days) are used to conduct objective synoptic circulation classification based on the T-mode PCA method with the Cost733 toolbox. Three levels of geopotential height fields (i.e., 925 850 and 500 hPa) in the lower to middle troposphere over 105º-125ºE and 30º-55ºN are used in circulation type (CT) classification. Six typical synoptic circulation types (CTs) are identified during winter in the BTH region, with a total explained variance of 70% (Fig. S1). The horizontal (i.e., sea level pressure (SLP), wind, relative humidity (RH) and boundary layer height (BLH)) and vertical (i.e., atmospheric stability, vertical velocity, temperature and divergence) distributions of meteorological variables are used to illustrate the mechanism behind CT effects on air pollution. To obtain a broad view of the six CTs, the horizontal distribution of atmospheric circulation patterns, as shown in Fig. 2 and Fig. 3 cover a larger area



than the area used in the CT classification with the Cost733 toolbox.
Fig. 2 and Fig. 3 exhibit the original and anomalous patterns of the mean SLP and surface wind field
of each CT, respectively. CT1 is the most frequent CT during the study period with an occurrence
frequency of 33% based on the results of the Cost733 classification. CT1 shows that a high-pressure
system originates in the Siberian region extending along central Inner Mongolia to southern China.
Northwesterly winds prevail in northern China and turn into northerly winds in southern China. The
mean wind speed is 3.27 m/s over the BTH region (cf. Table 2), which is the highest among the six
CTs and benefits the outward transport of local air pollutants. Fig. 3 shows the SLP and surface
wind anomalies of each CT. In the CT1 situation, the BTH region is located west of the cyclonic
anomaly, which is dominated by an obvious northwesterly wind anomaly. The wind field pattern
corresponds to the negative RH anomaly over the BTH region in Fig. 4. The vertical profiles of
dynamic and thermodynamic stratification are included to investigate vertical diffusion. Based on
the vertical distribution of atmospheric stability shown in Fig. 5, atmospheric stratification is
characterized by a stable layer at the top of the boundary layer for all the cases. For CT1, an obvious
unstable stratification occurs at the bottom of boundary layer over the BTH region, which enhances
the turbulent activities and is beneficial for the vertical diffusion of air pollutants. The unstable
boundary layer is also confirmed by the positive BLH anomaly and elevated negative temperature
anomaly, as shown in Fig. S2 and Fig. S3. Fig. S4 shows a strong surface divergence and strong top
convergence vertical pattern in CT1, which generates sinking movement over the BTH region. As
shown in Fig. 6, a subsidence anomaly appears at the lower to middle troposphere over the BTH
region with a mean descending velocity of 0.04 pa/s between 850 and 1000 hPa. The strong
downdraft brings a clean and dry air mass to the surface and increases the horizontal divergence of
surface air pollutants (shown in Fig. S4). The cold, clean and dry air mass transported by the surface
northwesterly winds, unstable boundary layer and strong horizontal divergence are favorable for the
improvement in ambient air quality.
The occurrence frequency of CT2 is 11%. As shown in Fig. 2, a high-pressure system around Baikal
is obvious under the CT2 condition, which is stronger and further east than CT1. The BTH region
is located at the ridge of the high-pressure system with weak northwesterly winds occurring in the



northern BTH region, which turn to northeasterly in the southern BTH region. The anomalous fields
in Fig. 3 show a large area of a positive SLP anomaly over the north of 40ºN. The BTH region is
just located at the south edge of the anticyclone anomaly with prevailing northeasterly surface wind.
Fig. 4 shows a weak negative RH anomaly over the BTH region due to the dry wind from the
northeast. Similar to CT1, CT2 also shows an unstable stratification in the boundary layer, which
increases the vertical diffusion of air pollution. Both the weak positive BLH anomaly and elevated
negative temperature anomaly indicate the enhanced instability of the atmospheric boundary layer
(Figs. S2-S3). Intense updraft is stimulated by strong convergence at the surface working with
strong divergence at the top of the boundary layer, as shown in Fig. S4. As shown in Fig. 6, upward
movement dominates in the middle-low troposphere over the BTH region with a mean ascending
velocity of 0.0358 pa/s between 850 and 1000 hPa. Although the elevated temperature stability is
relatively strong in CT2, the bottom-up updraft breaks through the stable layer and brings the surface
air pollutants to the free atmosphere. In summary, the unstable boundary layer working with the
upper divergence pumping action enhances the vertical diffusion of surface air pollutants, which
will decrease the surface concentrations of air pollutant.
CT3 shows a relatively uniform SLP distribution with a weak pressure gradient over the BTH region
as shown in Fig. 2. The prevailing westerly wind hinders the southward transport of the cold air
mass to some extent. The cyclonic anomaly with southwesterly wind can be found over the BTH
region. As shown in Fig. 3, the southwesterly wind transports the upstream air pollutants and warm
moisture to the BTH, which accelerates the hygroscopic growth of particles, promotes the gas-to-
particle transformation and increases the local air pollutant concentration (Wang et al., 2019a). The
positive RH and temperature anomaly in Fig. 4 and Fig. S3 correspond to the southwesterly wind
anomaly. Unlike to CT1 and CT2, CT3 shows a stable stratification below 700 hPa. In addition, the
upper unstable stratification of CT3 is lower than that of CT1 and CT2, indicating a negative BLH
anomaly (as shown in Fig. S2). CT3 also shows upward movement over the BTH region, but it is
weaker than CT2 by one order of magnitude. By contrast, the effects of the stronger near-surface
convergence will offset the upward transport, which will increase the local air pollutants. The stable
boundary layer, southeasterly warm moisture and effective convergence aggravate local air
pollution.



For the cases of CT4 and CT5, the BTH region is co-located with a weak surface anticyclone with
low average surface winds of 2.24 and 2.58 m/s, respectively. The calm surface winds coexisting
with the lower BLHs (cf. Fig. S2) decrease the ventilation coefficient and increase the occurrence
of air stagnation conditions. The surface anomaly fields show southeasterly and southerly winds in
CT4 and CT5, respectively. As shown in Fig. 4, the northward wind anomaly increases the humidity
and air pollutants of the BTH region. Based on the vertical profiles of temperature and atmospheric
stability, an elevated positive temperature anomaly increases the stability of the boundary layer, thus
reducing the vertical diffusion of air pollutants. The weak near surface convergence could increase
the accumulation of air pollution, but moderate upward movement will bring the surface air
pollutants to the outside of the boundary layer, which offsets the surface convergence to some extent.
CT4 and CT5 had the same occurrence of 15% during the study period. Although the CT4 and CT5
show different large-scale surface circulation patterns, the meteorological variables over the BTH
region are almost the same. The air stagnation conditions and southerly water vapor transport result
in the accumulation and hygroscopic growth of particles.
In terms of CT6, the BTH region is located at the ridge of the Mongolian anticyclone, and its high-
pressure system is weaker than that of CT2. The prevailing wind turns from northwest to northeast
over the BTH region. As shown by the surface meteorological anomaly distribution, the BTH region
is situated at the border between the northern anticyclonic and southern cyclonic anomalies with
prevailing northeasterly wind coming from the Bohai Sea. A large amount of water vapor from the
sea plays an important role in the hygroscopic growth of particles over the BTH region. Fig. 5
indicates a stable boundary layer when CT6 occurs, which reduces the vertical diffusion of surface
air pollutants. CT6 shows a deep horizontal convergence under 850 hPa, which is favorable for the
accumulation of moisture and air pollutants. The effect of the relatively weak divergence above
strong convergence is not distinct for the improvement in surface air quality. Therefore, the
circulation pattern of warm moist flow from the sea, a stable boundary and effective horizontal
convergence exacerbates local air pollution.
**3.2 Atmospheric circulation pattern effects on air quality**
The potential mechanisms of the CT effects on local air quality are discussed in the last section.





Combinations of the following situations are favorable for the improvement in air quality:
transport of a clean and dry air mass, unstable boundary layer, effective horizontal divergence and
vertical transport of air pollutants to the free atmosphere. In contrast, the positive humidity anomaly,
stable boundary layer, frequent air stagnation conditions and deep horizontal convergence
exacerbate air pollution.
To exclude the effects of interannual variation in air quality due to the emission reduction
background, the daily $PM_{2.5}$ concentration distribution displayed by year and CT, as shown in Fig.
7 reveals the effects of CT on air quality. The mean and median values of $PM_{2.5}$ concentrations
during each CT are summarized in Table 1. The mean and median $PM_{2.5}$ concentrations in the CT1
condition are both lower than the seasonal mean and median for all years. Under the CT2 condition,
the $PM_{2.5}$ concentrations are also lower than the seasonal mean except for 2014. However, the $PM_{2.5}$
concentrations are generally higher than the seasonal mean in CT3-CT6. As for the multiyear
average, it shows distinctly lower $PM_{2.5}$ concentrations in CT1 and CT2 than the other CTs. Based
on the $PM_{2.5}$ concentration in each CT, CT1 and CT2 can be considered as favorable CTs for air
quality, which are beneficial for the diffusion of air pollutants, and CT3-CT6 are unfavorable CTs,
which exacerbate air pollution.
Giving the above analysis, $PM_{2.5}$ concentration tended to be lower than normal when a favorable
CT occurred, and vice versa. Therefore, the occurrence frequency of each CT plays an important
role in air quality during the study period. CT1 and CT2 are combined as the favorable circulation,
and CT3-CT6 are referred to as the unfavorable circulation. Fig. S5 exhibits the seasonal
occurrences of favorable and unfavorable circulation types. Fifty-four days of unfavorable
circulation occurred in winter 2013, which is the greatest frequency during the study period. A
higher unfavorable circulation frequency was also shown in 2014 and 2018 winters. In contrast, the
favorable circulations were much higher in 2015 and 2017 winters than in the other winters. The
seasonal frequencies of favorable and unfavorable circulations are in line with the trend in seasonal
$PM_{2.5}$ concentrations. It is worth noting that although the seasonal mean $PM_{2.5}$ concentration in the
winter of 2015 (Dec. 2015 to Feb. 2016) is lower than that of 2014, but the $PM_{2.5}$ concentration in
Dec. 2015 is much higher than that in Dec. 2014. The high $PM_{2.5}$ concentration in Dec. 2015 is





consistent with the high frequency of unfavorable CTs during that time, which indicates the
robustness of circulation classification.
However, every air pollution event has a duration from the development to decay stage. Generally,
several days are needed for the accumulation of air pollutants, followed by a relatively quick
removal. The variation in meteorological conditions controls the evolution of each air pollution
episode. Therefore, the duration of each CT determines the duration of the air pollution event. Fig.
8 exhibits the variation in the $PM_{2.5}$ concentration anomaly with the duration of favorable and
unfavorable CTs. As discussed above, the favorable circulations generally correspond to the
negative $PM_{2.5}$ concentration anomaly (lower than the monthly mean), while the unfavorable
circulations result in a positive $PM_{2.5}$ concentration anomaly. When the favorable circulation
duration shorter than 4 days, the absolute values of the negative anomaly of $PM_{2.5}$ concentrations
increase with the duration of favorable circulation; however, with the continuous increase in
favorable circulation durations, the magnitude of the negative anomaly of $PM_{2.5}$ concentrations
slightly decreases and remains unchanged. Similarly, the positive anomalies of the $PM_{2.5}$
concentrations increase with the duration of unfavorable circulation durations when the duration is
less than 7 days. However, the effect of circulation on air pollutant diffusion is not obvious when a
one-day favorable or one-two-day unfavorable circulation occurs. That is favorable CTs lasting 2~4
days are beneficial for the diffusion of air pollutants; and unfavorable circulation events lasting 3~7
days exacerbate the accumulation of air pollutants.
The occurrences of 2~4 days favorable circulation and 3~7 days of unfavorable CTs are shown in
Fig. 9. It shows a high frequency of 2~4 days of favorable circulation in 2017 and 2014 with totally
15 and 13 days, respectively. The favorable circulation occurrences are lower in the winters of 2016
and 2018 than in the other winters. In terms of the 3~7 days of unfavorable circulations, the years
of 2013, 2016 and 2018 show higher frequencies than the other years. Therefore, based on the
occurrence of favorable and unfavorable CTs, the atmospheric diffusion abilities are better in 2014
and 2017 than in the other years. The significant improvement in air quality in 2014 and 2017 is
consistent with the improvement in atmospheric diffusion abilities compared to their previous years.





**3.3 Contributions of atmospheric diffusion condition variations to the PM$_{2.5}$ concentration**
**decrease between 2016 and 2017**
Although the interannual variation in PM$_{2.5}$ concentrations show good correlation with the
occurrence of favorable or unfavorable circulation, Sec. 3.2 is just a qualitative analysis. Taking the
interannual variation in PM$_{2.5}$ concentrations between 2016 and 2017 as an example, the model
simulation based on the WRF-Chem model is used to evaluate the quantitative contributions of
meteorological condition variations to the PM$_{2.5}$ concentration decrease in 2017. The emissions are
fixed in 2016 (Dec. 2016 to Feb. 2017), and the meteorological fields come from the NECP GDAS
Final Analysis dataset for the 2016 and 2017 winters, respectively. The meteorological fields and
air pollutants over some cities from north to south in the simulated domain (i.e., Shijiazhuang,
Beijing, Tianjin, Xuzhou and Shanghai) are included to evaluate the performance of the model
simulation. Fig. S6 shows the variations in the observed and simulated daily mean air temperature,
sea level pressure, relative humidity and surface wind speed from Jan. to Feb. of 2017. Although
the model slightly overestimates the surface wind speed over Shijiazhuang and Shanghai, most of
the simulated meteorological variables agree well with the observations over all cities. For the
concentration of air pollutants in Fig. S7, the model generally underestimates the PM$_{2.5}$
concentrations under highly polluted conditions, with a bias of 44.9%~59.6% (different cities) when
the observed PM$_{2.5}$ was higher than 75 μg/m³. However, the bias between the simulated and observed
PM$_{2.5}$ concentrations decreased to 12.4%~26.8% at lower PM$_{2.5}$ concentration level. Due to the
deficiency of the PBL scheme (Tie et al., 2015), the heterogeneous/aqueous process in the model
(Li et al., 2011) and uncertainty in the emission inventory, current air quality models show limited
capacity in severe air pollution episodes. However, the day-to-day variation in all the air pollutants
can be well captured by the WRF-Chem model, with the highest correlation coefficient of 0.76
between the observed and simulated PM$_{2.5}$ in Xuzhou. Overall, both the meteorological variables
and air pollutants are well reproduced by the WRF-Chem model, which provides confidence for
further discussions.
The simulated seasonal mean PM$_{2.5}$ concentrations of the 2016 and 2017 winters are presented in
Fig. S8. It shows a significant spatial distribution of seasonal PM$_{2.5}$ concentrations with higher



concentrations over the BTH region, Shandong and Henan Provinces. Even though the emissions
were set to the level of 2016, the simulated seasonal PM$_{2.5}$ concentrations in 2016 were much higher
than those in 2017 due to the difference in meteorological fields. Fig. 10 exhibits the observed and
simulated PM$_{2.5}$ concentration differences between 2017 and 2016. Both the observations and
simulations show significant negative growth in PM$_{2.5}$ concentrations over northern China from
2016 to 2017 in winter but relatively weak positive growth over the lower Yangtze River Delta. The
BTH region is located at the center of negative growth, with an observed average of 46.3 μg/m$^3$
decrease of PM$_{2.5}$ concentration from 2016 to 2017. While, the simulated difference of PM$_{2.5}$
between 2016 and 2017 winter is -8.4 μg/m$^3$, which is much lower than the observed value. The
absolute PM$_{2.5}$ concentration would be underestimated because of the limited performance of the
WRF-Chem model under severe air pollution; therefore, the relative differences between 2016 and
2017 are involved to evaluate the effects of meteorological field variations on the decrease in PM$_{2.5}$
concentrations. Based on the relative difference in PM$_{2.5}$ concentration between 2016 and 2017, the
observed difference at 114 stations over the BTH region is -37.1% compared to the mean value of
2016 winter, and the averaged simulated difference is -28.4% over the region of 113º-117.5ºE and
36º-42ºN, which is due to the difference in meteorological conditions. Thus, 76.5% of the observed
37.1% decrease in PM$_{2.5}$ concentration in 2017 over the BTH region could be attributed to the
improvement in atmospheric diffusion conditions. The variation of meteorological conditions plays
an important role in the interannual variation in air pollutant concentrations.

## 4. Conclusions and Discussion

Recent severe PM$_{2.5}$ pollution in China has aroused unprecedented public concern. The Chinese
government has implemented many emission reduction measurements, which has greatly improved
the air quality recently. The wintertime PM$_{2.5}$ concentration of 2018 decreased by 35.6% compared
to 2013 over the BTH region. However, there was obvious interannual variation in PM$_{2.5}$
concentrations from 2013 to 2018. Atmospheric circulation classification method based on the
Cost733 toolbox is used to investigate the mechanism behind atmospheric circulation effects on air
pollutant diffusion. Six CTs are identified during the winters from 2013 to 2018 over northern China,



and two of which are considered as favorable circulations for air pollutant diffusion and the other
four CTs exacerbate local air pollution. Generally, the transport of clean and dry air mass and
unstable boundary layers working with the effective near-surface horizontal divergence or pumping
action at the top of the boundary layer will benefit for the horizontal or vertical diffusion of surface
air pollutants. However, the co-occurrence of a stable boundary layer, frequent air stagnation,
positive water vapor advection and deep near-surface horizontal convergence exacerbates the air
pollution.
Except for the atmospheric circulation characteristic of CTs, the durations of each circulation type
also have a great influence on the local air quality. The one-day favorable or less than two-day
unfavorable circulations have no significant effects on the diffusion and accumulation of air
pollutants. Comparatively speaking, favorable CTs lasting for 2~4 days are beneficial for the
diffusion of air pollutants, and the 3~7 days of unfavorable circulation events exacerbate the
accumulation of air pollutants. The occurrences of 2~4 days of favorable and 3~7 days of
unfavorable circulation are used to evaluate the atmospheric diffusion ability, which shows better
diffusion abilities in 2014 and 2017 than in the other years. Taking the decrease of $PM_{2.5}$
concentration between 2016 and 2017 as an example, 76.5% of the decreased concentration over
the BTH region could be attributed to the improvement in atmospheric diffusion conditions of 2017.
The variation in meteorological conditions plays an important role in the interannual variation in air
pollutant concentrations. The 2020 is the key and target year for the three-year action to win the
battle for a blue sky of 2018. It is essential to exclude the contribution of meteorological conditions
to the variation in interannual air pollutants when making a quantitative evaluation of emission
reduction measurements.

**Acknowledgments:** This study was supported by the National Natural Science Foundation of China
(41790470 and 41805117).

**Code/Data availability:** The release version 4.0 of WRF-Chem can be download from
http://www2.mmm.ucar.edu/wrf/users/download/get_source.html. Hourly $PM_{2.5}$ concentration


observations were obtained from the website of Ministry of Ecology and Environment of the
People's Republic of China (http://106.37.208.233:20035). Daily four times ECMWF ERA5 dataset
during 2013 to 2018 are downloaded from https://www.ecmwf.int/en/forecasts/datasets/reanalysis-
datasets/era5. Hourly observations of meteorological variables used for the WRF-Chem simulation
evaluations are downloaded from the Intergrated Surface Database of National Climate Data Center
(https://www.ncdc.noaa.gov/isd).

**Competing interests:** The authors declare that they have no conflict of interest.

**Author contributions:** Wang X. and Zhang R. designed research; Wang X. performed the analyses
and wrote the paper; All authors contributed to the final version of the paper.



**Figure Captions:**


Figure 1. Interannual variation in the wintertime $PM_{2.5}$ concentrations at 114 stations over the BTH region. In each
box, the central mark indicates the median, and the bottom and top edges of the box indicate the 25th and 75th
percentiles, respectively. The whiskers extending to the most extreme data points are considered outliers. The region
covered by the blue box in Fig. 2 is considered as the BTH region (113º-117.5ºE and 36º-42ºN).
Figure 2. The distribution of sea level pressure (shaded, unit: pa) and 10 m wind fields (vector, unit: m/s) in each
circulation type. The number over each subplot indicates the occurrence frequency of the specific circulation type.
The solid blue box is the location of BTH region. The daily mean geopotential height fields at 925, 850 and 500 hPa
over the dashed blue box (105º-125ºE and 30º-55ºN) were applied to T-mode PCA method with the cost733 toolbox.
The region mean wind speed of each circulation type is shown in Table 2.
Figure 3. The distribution of sea level pressure (unit: pa) and 10 m wind fields (unit: m/s) anomaly in each circulation
type. The anomaly values are with respect to the 1980-2010 mean. Regional mean wind speed anomaly of each
circulation type is summarized in Table 2.
Figure 4. The distribution of relative humidity in each circulation type (unit: %). The anomaly values are with respect
to the 1980-2010 mean.
Figure 5. Zonal profile of temperature lapse rate over the BTH region (36º-42ºN) (unit: K/100 m). The gray region
indicates the average altitude over 36º-42ºN. The region between the two dashed lines is the horizontal location of
the BTH region (113º-117.5ºE).
Figure 6. Zonal vertical profile of vertical velocity anomaly over BTH region (unit: pa/s). The anomaly of the vertical
velocities is with respect to the 1980 to 2010 mean value.
Figure 7. The box plot of the $PM_{2.5}$ concentrations varies with the circulation types. To exclude the effect of emission
reduction on the annual mean $PM_{2.5}$ concentrations, the $PM_{2.5}$ distributions at the year and multiyear (average) scales
are shown here, respectively. The dashed line for each year indicates the median $PM_{2.5}$ concentrations in wintertime
of a specific year.
Figure 8. The daily $PM_{2.5}$ concentration anomalies vary with favorable (F) and unfavorable (U) event durations. The
occurrences of CT1 and CT2 are collectively called favorable events, and CT3 to CT6 are referred to as unfavorable
events. U1 indicates an unfavorable circulation event lasting for one day, and U2 means a two-day event. The central
red line in each box indicates the median, and the circle is the mean value.
Figure 9. Occurrence frequencies of the effective favorable and unfavorable events. The effective favorable events
referred to the favorable events lasting for two to four days. The effective unfavorable events indicate the unfavorable





events lasting for three to seven days. The specific number of days for favorable/unfavorable events is shown on the
top of each bar.
Figure 10. Distributions of the observed and simulated PM$_{2.5}$ difference between the winters of 2016 and 2017. The
left panel is the absolute value (unit: μg/m$^3$) and the right panel is the relative difference with respect to the mean
value of 2016 (unit: %). The simulated seasonal mean PM$_{2.5}$ concentrations during the two years are shown in Fig.
S8.





Table 1. The seasonal mean and median PM₂.₅ concentrations in each atmospheric circulation type (CT) over the
BTH region. PM₂.₅ concentrations in bold represent the mean/median value of each CT lower than the all-case
seasonal mean/median value.

| Seasonal Mean/ Median ($\mu g/m^3$) | CT1 | CT2 | CT3 | CT4 | CT5 | CT6 |
|---|---|---|---|---|---|---|
| 2013 (123.97/97.23) | **104.99/71.42** | **94.51/69.33** | 144.76/118.50 | 135.47/117.20 | 166.28/156.52 | **67.90/47.21** |
| 2014 (93.07/75.79) | **71.03/51.52** | 122.99/109.37 | 105.91/96.82 | **86.26/72.06** | 115.37/94.69 | 118.16/110.17 |
| 2015 (95.67/65.97) | **58.56/38** | **89.38/73.07** | 134.77/114.69 | 135.91/106.36 | 124.15/99.81 | 106.14/70.63 |
| 2016 (112.94/91.32) | **84.74/66.16** | **110.02/88.10** | 138.96/114.26 | 122.86/95.02 | 142.52/128.77 | 132.95/129.52 |
| 2017 (70.44/54.07) | **56.49/43.16** | **60.70/39.61** | 80.03/67.39 | 83.89/67.24 | 93.63/79.28 | **69.77/52.23** |
| 2018 (79.85/63.02) | **77.99/60.68** | **51.77/37.43** | 89.26/77.57 | 86.70/81.35 | **75.08/52.72** | 108.60/93.02 |
| AVERAGE (95.27/72.22) | **73.14/53.04** | **79.12/54.89** | 115.18/96.29 | 109.85/88.25 | 116.04/89.04 | 100.40/82.04 |


Table 2. Regional mean meteorological variables over the BTH region under each circulation type

| Variables | CT1 | CT2 | CT3 | CT4 | CT5 | CT6 |
|---|---|---|---|---|---|---|
| Surface wind speed (m/s) | 3.27 | 2.31 | 2.71 | 2.24 | 2.58 | 2.54 |
| Surface wind speed anomaly (m/s) | 0.53 | -0.42 | -0.04 | -0.49 | -0.15 | -0.19 |
| Mean vertical velocity anomaly between 850 to 1000 hPa (pa/s) | 0.04 | -0.0358 | -0.0038 | -0.0296 | -0.0111 | -0.0213 |
| Difference of temperature anomaly between 850 and 1000 hPa (K) | -0.716 | -0.206 | 0.664 | 0.456 | 0.232 | 0.485 |


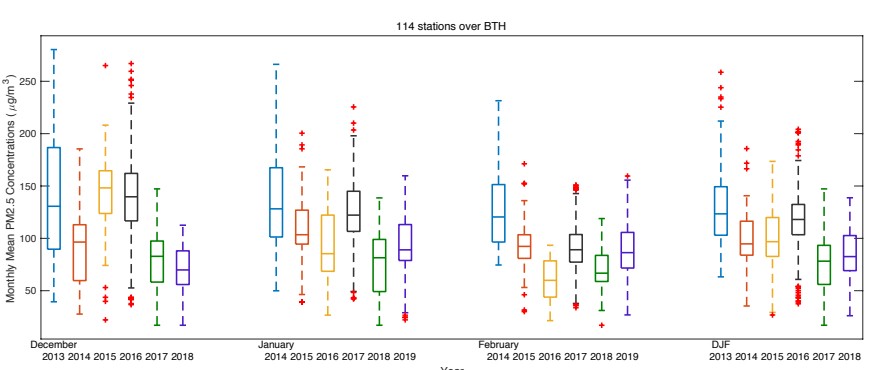


Figure 1. Interannual variation in the wintertime PM$_{2.5}$ concentrations at 114 stations over the BTH region. In each

box, the central mark indicates the median, and the bottom and top edges of the box indicate the 25th and 75th

percentiles, respectively. The whiskers extending to the most extreme data points are considered outliers. The region

covered by the blue box in Fig. 2 is considered as the BTH region (113º-117.5ºE and 36º-42ºN).

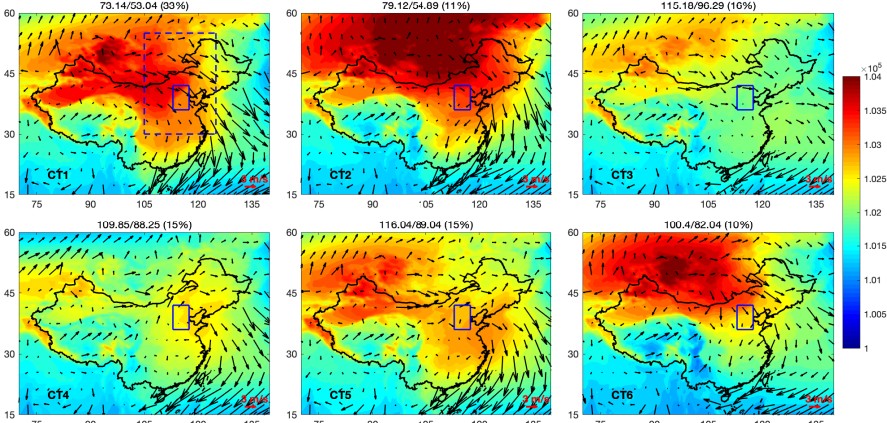

473

Figure 2. The distribution of sea level pressure (shaded, unit: pa) and 10 m wind fields (vector, unit: m/s) in each

circulation type. The number over each subplot indicates the occurrence frequency of the specific circulation type.

The solid blue box is the location of BTH region. The daily mean geopotential height fields at 925, 850 and 500 hPa

over the dashed blue box (105º-125ºE and 30º-55ºN) were applied to T-mode PCA method with the cost733 toolbox.

The region mean wind speed of each circulation type is shown in Table 2.



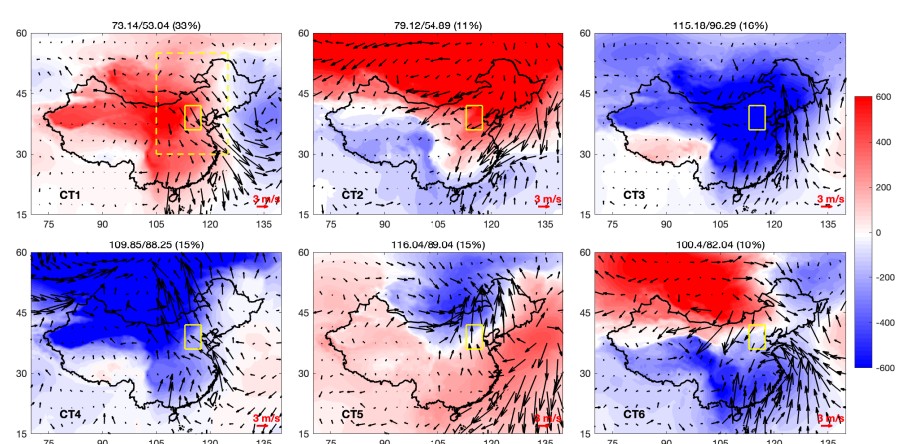

Figure 3. The distribution of sea level pressure (unit: pa) and 10 m wind fields (unit: m/s) anomaly in each circulation
type. The anomaly values are with respect to the 1980-2010 mean. Regional mean wind speed anomaly of each
circulation type is summarized in Table 2.

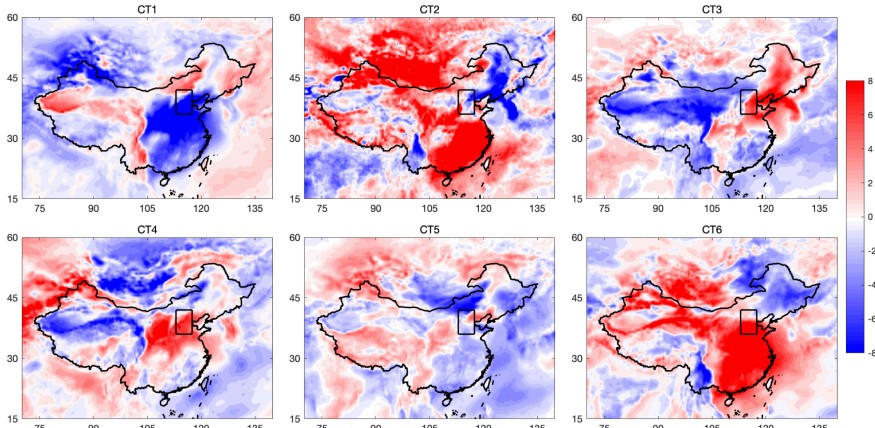

Figure 4. The distribution of relative humidity in each circulation type (unit: %). The anomaly values are with respect
to the 1980-2010 mean.


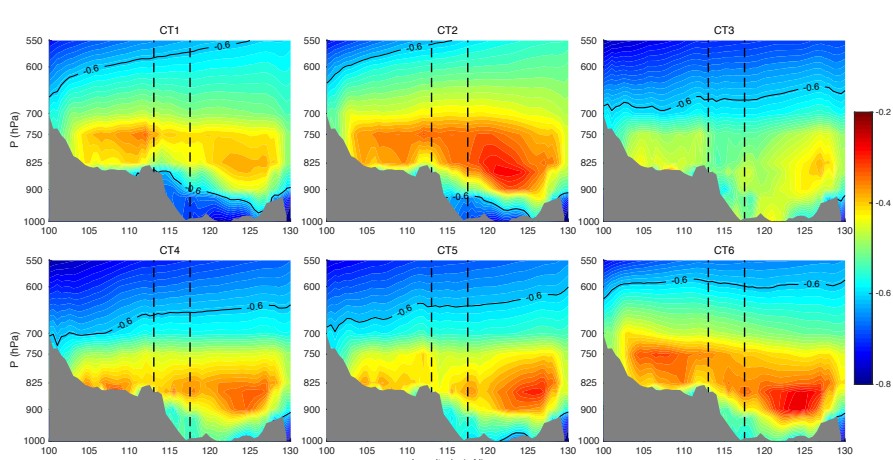

Figure 5. Zonal profile of temperature lapse rate over the BTH region (36º-42ºN) (unit: K/100 m). The gray region indicates the average altitude over 36º-42ºN. The region between the two dashed lines is the horizontal location of the BTH region (113º-117.5ºE).

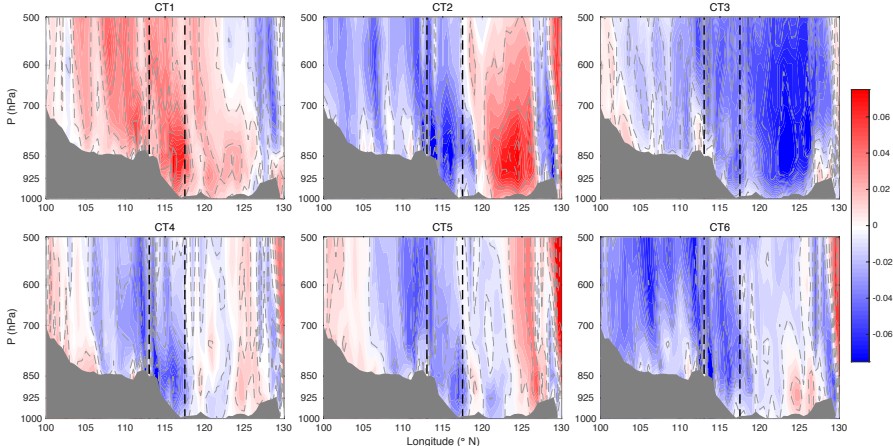

Figure 6. Zonal vertical profile of vertical velocity anomaly over BTH region (unit: pa/s). The anomaly of the vertical velocities is with respect to the 1980 to 2010 mean value.

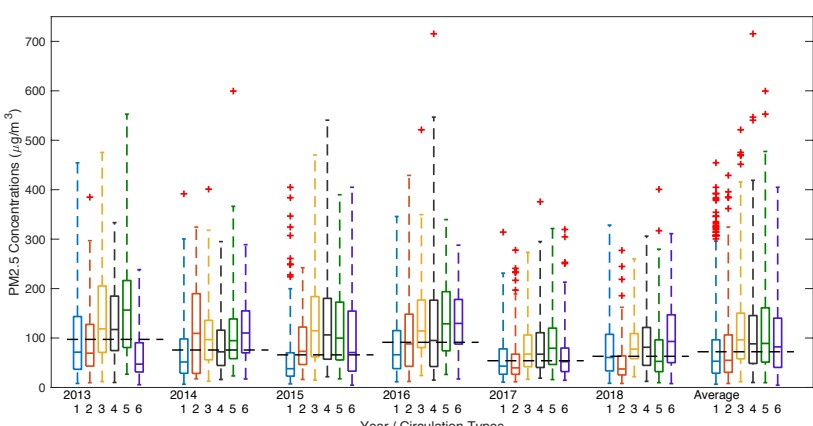

496

Figure 7. The box plot of the PM$_{2.5}$ concentrations varies with the circulation types. To exclude the effect of emission
reduction on the annual mean PM$_{2.5}$ concentrations, the PM$_{2.5}$ distributions at the year and multiyear (average) scales
are shown here, respectively. The dashed line for each year indicates the median PM$_{2.5}$ concentrations in wintertime
of a specific year.

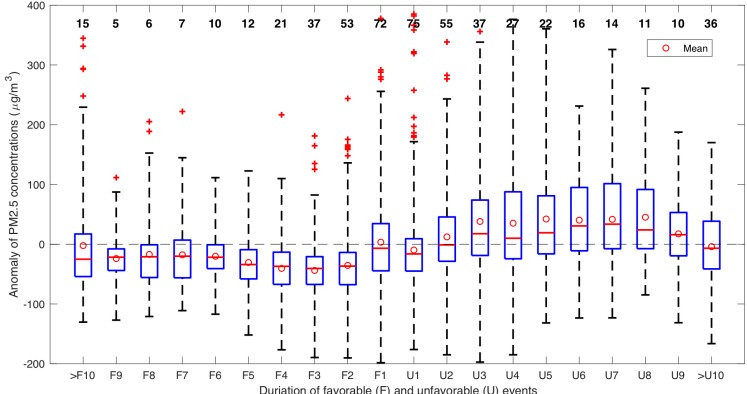

501

Figure 8. The daily PM$_{2.5}$ concentration anomalies vary with favorable (F) and unfavorable (U) event durations. The
occurrences of CT1 and CT2 are collectively called favorable events, and CT3 to CT6 are referred to as unfavorable
events. U1 indicates an unfavorable circulation event lasting for one day, and U2 means a two-day event. The central
red line in each box indicates the median, and the circle is the mean value.

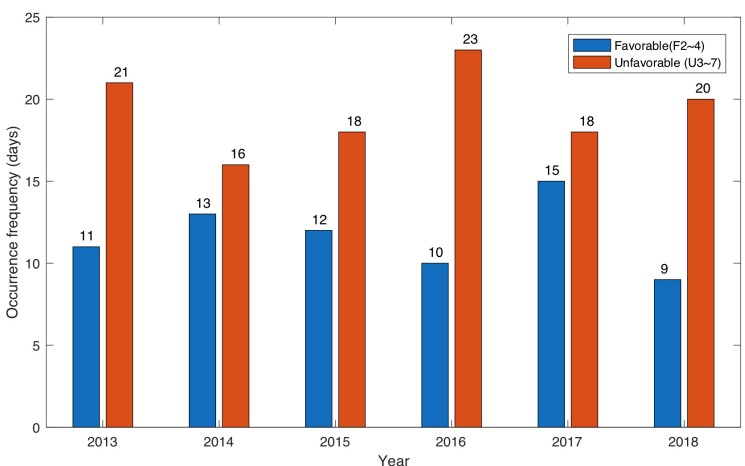


Figure 9. Occurrence frequencies of the effective favorable and unfavorable events. The effective favorable events
referred to the favorable events lasting for two to four days. The effective unfavorable events indicate the unfavorable
events lasting for three to seven days. The specific number of days for favorable/unfavorable events is shown on the
top of each bar.

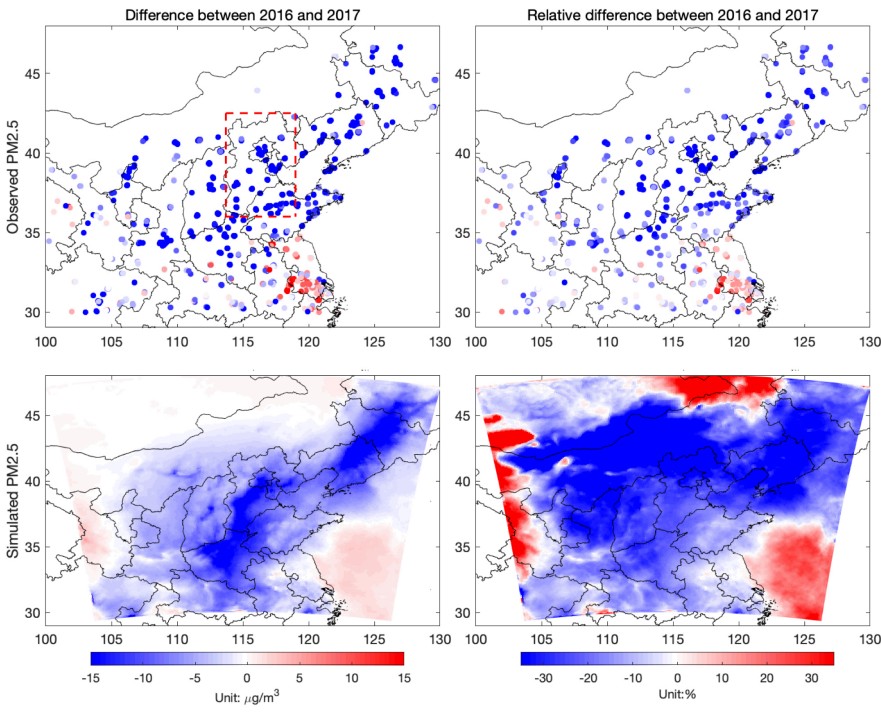


Figure 10. Distributions of the observed and simulated PM$_{2.5}$ difference between the winters of 2016 and 2017. The





left panel is the absolute value (unit: $\mu g/m^3$) and the right panel is the relative difference with respect to the mean
value of 2016 (unit: %). The simulated seasonal mean $PM_{2.5}$ concentrations during the two years are shown in Fig.
S8.



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
