# Peer review of "Effects of atmospheric circulations on the interannual variation in"

_Atmospheric Chemistry and Physics, 2020_

## Referee Comment (RC1) · Anonymous Referee #1 · 19 Mar 2020

Review of "Effects of atmospheric circulations on the interannual variation in PM2.5 concentrations over the Beijing-Tianjing-Hebei region in 2013-2018" by Wang and Zhang (MS ID#ACP-2020-198)

This study was aimed to explore the possible contribution of atmospheric circulation anomaly on the interannual variation of winter PM2.5 over northern China. Six dominate synoptic circulation types that favorable and unfavorable for the PM2.5 diffusion are revealed, which is interesting and quite important for us. Furthermore, the authors revealed that there is approximately 76.5% of the observed decrease in PM2.5 concentrations in 2017 over BTH could be attributed to the improvement of the atmospheric diffusion conditions. This paper is well written and organized, and there is no big flaw. I recommend it to be published in ACP after several minor corrections.

1. In this study, the authors have explored that there is approximately 76.5% of the observed decrease in PM2.5 could be attributed to the improvement of the atmospheric diffusion conditions. That is, the contribution of effect of atmospheric anomaly exceeded 70%, which presented far larger than that from the early studies and also confused me. As description in Introduction, the effect of atmospheric anomaly was just accounting for about 5% or 12%. Is there any idea about this large difference? Moreover, the additional discussion about the uncertainty of the evaluated contribution should be added. Is it related to the large bias of the WRF-CHEM model? 2. The winter season should be highlighted in the abstract. 3. More detailed introduction about the rotated T-mode PCA method was suggested. 4. The synoptic types of CT1 and CT2 is favorable for the air pollution divergence, while CT3-CT6 is unfavorable. CT3-CT6 can account for 56% of the weather types. How about it from the WRF-CHEM model? 5. How about the atmospheric circulation patterns in year 2016? The PM2.5 in this year was recovered and higher than the other years. How large contribution of the atmospheric circulation effect in your mind? Or the high PM2.5 is mainly sourced from the emission.

---

## Referee Comment (RC2) · Anonymous Referee #2 · 20 Mar 2020

This study makes a full investigation about the effects of atmospheric circulations on the interannual variation of PM2.5 over the Jing-Jin-Ji region, which is interesting and valuable to both science community and the society. It defines six types of atmospheric circulations and reveals their roles (favor or unfavor) to the PM2.5 concentration. In principle, the paper is a good contribution to the science community and worthy for publication after a minor revision as suggested below.

General Comments (1)It is always a puzzle regarding the relative contribution from emissions, meteorology, climate and topography to the aerosols observed. The authors

found the results in Line 365-375, regarding which I have two questions here. The first one, also the most important one, the relative contribution from meteorological contribution found here is ~37% for most stations, which is much higher than the values found by other studies (~10%), then why? May you please give an explanation? The second one, with the same emission map, the decrease of PM2.5 from simulations between 2017 and 2016 is larger than that from observation, which seems to me that it implies more emissions in 2017 than in 2016. Is this true or possible? (2)Another thing is that the aerosol pollution is often coupled with the meteorology, causing non-linear relationships between aerosol pollution and meteorology or aerosol emissions. In other words, the relative contribution from both factors could vary with the air pollution cases. How could you account for this coupled effect?

Detailed Comments Line 17 and 43, full spell should be provided for PM2.5 when first used. Line 44-47, regarding the aerosol effect on climate by changing the surface radiation balance, four more references are recommended, Garrett and Zhao (2006, DOI: 10.1038/nature04636) and Zhao and Garrett (2015, doi:10.1002/2014GL062015) showed the aerosol's strong warming effect in the winter Arctic through increasing cloud thermal emissivity; Zhao et al. (2020, https://doi.org/10.1093/nsr/nwz184) showed the impacts of aerosols on the weather and climate by changing the radiation over the Tibetan Plateau; and Yang et al. (2018, DOI: 10.1016/j.atmosres.2018.04.029) showed the cooling effect of aerosols to Hongkong region climate during past 30 years. Line 47, "pollutions" -> "Pollution" Line 42, "air quality" -> "the air quality" Line 51-52, you may change the second "strengthening" to "improving" Line 64-67, One more reference could be also cited, which show significant improvement of air quality in five typical cities in China during recent several years, along with detailed discussions about the potential reasons for pollutions in these cities, Zhang et al. (2019, https://doi.org/10.1007/s13143-019-00125-w). Line 69-72, these are true, which also include the dilution due to increasing the planetary boundary layer (Yang et al., 2016, doi:10.1002/2015JD024645), exchange of polluted and clean air, and hygroscopic growth of aerosols (Sun et al., 2019, DOI: 10.1029/2019EA000717;

Zhao et al. 2018, https://doi.org/10.1007/s00376-017-7069-3). Moreover, Garrett et al. (2010, https://doi.org/10.1111/j.1600-0889.2010.00453.x) demonstrates the importance of long-range transport and wet scavenging to the aerosol amount in the Arctic; Sun et al. (2019) showed the relative roles of wet scavenging and hygroscopic growth the aerosols in Beijing, and Zhao et al. (2018) showed the fast growth of fine aerosols particles in Beijing. Line 72-76, regarding the climate signals, this is important. One thing I am not sure is how to differ this with short-term meteorological influence. The other is that Chen et al. (2019, https://doi.org/10.1007/s00382-019-04706-3) suggested that the Arctic warming have a strong tele-connection with mid-latitude air pollution (aerosol amount). For example, an increase in Arctic surface temperature in summer is associated with enhanced air pollution in Asia in winter. Line 80, "contribution to" -> "contribution from" Line 106-108, since you are focusing on the winter time PM2.5 mass concentration over the BTH region, I would suggest to add a short paragraph to describe the wintertime PM2.5 pollution over the BTH region in the introduction part. Line 115, the region is also defined in Figure 2, why not using Figure 2? Line 116-117, Does this imply that the daily data is set as missing when the missing data is more than 40% in a day? Line 119-120, what do you mean "nonlinear methods" here? Line 125, "Zhang et al. (Zhang et al., 2012)" -> "Zhang et al. (2012)" Line 126-127, what do COST and PM mean here? Line 139, NCEP FNL should be fully spelled when first used. Line 151-153, This is true. However, it just represents one case with different meteorology (2016 vs 2017). You may add one sentence to assume that this result is used to represent the typical value of meteorological contribution to PM2.5 concentration. Line 157-158, Why do you use so long time to spin-up (15 days)? May you please briefly explain? Line 163, "Dominate" -> "Dominant" Line 177-178, "the accumulate" -> "accumulate". In other word, remove "the". Line 273, "in the last section": do you mean "this section"? Line 299-301, delete either "although" or "but" Line 311-313, "when the favorable circulation duration shorter . . ." -> "when the favorable circulation duration is shorter . . ." Line 365-375, I have two questions here. The first one, also the most important one, the relative contribution from meteorological

contribution found here is ∼37% for most stations, which is much higher than the values found by other studies (∼10%), then why? May you please give an explanation? The second one, with the same emission map, the decrease of PM2.5 from simulations between 2017 and 2016 is larger than that from observation, which seems to me that it implies more emissions in 2017 than in 2016. Is this true or possible? Line 403-404, I would suggest "The 2020 is the key and target year for the three-year action to win the battle for a blue sky goal set in 2018".

---

## Short Comment (SC1) · 27 Apr 2020

**Review 1**

**Comment:** This study was aimed to explore the possible contribution of atmospheric circulation anomaly on the interannual variation of winter PM2.5 over northern China. Six dominate synoptic circulation types that favorable and unfavorable for the PM2.5 diffusion are revealed, which is interesting and quite important for us. Furthermore, the authors revealed that there is approximately 76.5% of the observed decrease in PM2.5 concentrations in 2017 over BTH could be attributed to the improvement of the atmospheric diffusion conditions. This paper is well written and organized, and there is no big flaw. I recommend it to be published in ACP after several minor corrections.

**Response:** Thank you very much for the through and helpful comments and suggestions. Please find the following point-point response.

**General comments:**

**Comment 1:** In this study, the authors have explored that there is approximately 76.5% of the observed decrease in PM2.5 could be attributed to the improvement of the atmospheric diffusion conditions. That is, the contribution of effect of atmospheric anomaly exceeded 70%, which presented far larger than that from the early studies and also confused me. As description in Introduction, the effect of atmospheric anomaly was just accounting for about 5% or 12%. Is there any idea about this large difference? Moreover, the additional discussion about the uncertainty of the evaluated contribution should be added. Is it related to the large bias of the WRF-CHEM model?

**Response 1:** Chinese government issued the Clean Air Action in 2013 to mitigate PM2.5 pollution. Most of the existing researches we involved in Introduction are focused on the evaluation of Clean Air Action from 2013 to 2017 or 2018. During the five to six years, the average contribution of meteorological conditions to the air quality improvement is assessed as 5% or 12% depends on different methods and domains. The primary concern of our paper is to investigate the effects of meteorological elements on the interannual variation of air quality, the magnitude of which may be larger than the multiyear averaged value. Moreover, based on the occurrence of different circulation types in Fig. 9, 2016 and 2017 winters are the most unfavorable and the most favorable diffusion conditions during the study period, respectively, which may be the reason for the significant and high contribution of meteorological factor in our result. In addition, the observed PM2.5 variation average between 2016 and 2017 was calculated based on

the PM2.5 observations at 114 stations, while, the simulated PM2.5 differences are derived from the grid results over the region of 113º-117.5ºE and 36º-42ºN in our original version. However, both observed and simulated PM2.5 different between 2016 and 2017 show obvious spatial distribution in Fig. 10. To exclude the effects of spatial distribution, the simulated grid results are interpolated to PM2.5 observation stations in the revised version. The simulated PM2.5 difference between 2016 and 2017 reduced from the original 28.4% to current 22.6%, and the relative contribution rate of meteorological elements is 60% from 2016 to 2017 winter. That is to say, 40% of the 37.7% (i.e., 15%) reduction in PM2.5 concentration can be attributed to the emission reduction between the two consecutive years. It is generally known that one of the goals of Clean Air Action is to decrease PM2.5 concentrations by 25% in Jing-Jin-Ji regions from 2013 to 2017. Based on our simulation, the 15% reduction of emission from 2016 to 2017 accounts for large part of the overall target of 2013 to 2017, which verified the robust of the relative 60% contribution of meteorological elements during the selected two consecutive years.

Some discussions about the uncertainty of WRF-Chem simulation are added in Lines 431-439: *The quantitative evaluation of meteorological elements contribution to the interannual variation of PM2.5 concentrations between winters of 2016 and 2017 is derived from the WRF-Chem simulation in this study. Although the model performance for PM2.5 is generally satisfactory in Fig. S7, it shows obvious underestimation in the severe haze days. Reasons for these biases might be the overestimation in surface wind speed, uncertainties of emission inventory and insufficient treatments of some new chemistry mechanisms of particle formation, which need be further discussed in the future. In addition, some emission modules are turned off to reduce the computation cost, i.e., dust, sea salt, dimethyl sulphide, biomass burning and wildfires, which would result in the uncertainty of simulated PM2.5 mass concentrations.*

**Comment 2:** The winter season should be highlighted in the abstract.

**Response 2:** We clarify the wintertime as the study period in the abstract and introduction sections.

**Comment 3:** More detailed introduction about the rotated T-mode PCA method was suggested.

**Response 3:** More detailed information about T-mode PCA is involved to further improve the method of atmospheric circulation classification in Lines 133-138 and Lines 149-152.

Lines 133-138: *In this model, the input data matrix is space-time two-dimensional: the rows represent spatial grids, and the columns is time series. The data are divided into ten subsets to speed up computations, and the principal components (PCs) are achieved using the singular value decomposition for each subset and an oblique rotation is applied to the PCs to achieve better classification effects. Then, chi-square test is used to evaluate the ten classifications based on the subsets and the subset with the highest sum is chosen and assigned to a type.*

Lines 149-152: *Prior to using Cost733, the number of principal components need to be defined manually. To exclude the influences of various number of principal components, sensitivity tests with principal components from 2 to 10 are conducted in this study, the explained variances of which are shown in Fig. S1.*

**Comment 4:** The synoptic types of CT1 and CT2 is favorable for the air pollution divergence, while CT3-CT6 is unfavorable. CT3-CT6 can account for 56% of the weather types. How about it from the WRF-CHEM model?

**Response 4:** The occurrence of CT3-CT6 is 56% throughout the study period, which may be different in the specific year. The circulation classification can be considered as a semiquantitative method to evaluate the capacity of air pollution diffusion, but the explained variances of classifications is 70% as show in Fig. S1, which indicates some uncertainty of the method. To give a quantitative assessment of meteorological elements contribution to the air quality improvement, distribution of air pollutants and meteorological condition in winters of 2016 and 2017 are simulated in our work. We evaluated the performance of simulated meteorological fields based on the station observed daily mean wind speed, temperature, pressure and relative humidity in Fig. S6, which is more quantitative than the occurrence frequency of circulation classifications.

**Comment 5:** How about the atmospheric circulation patterns in year 2016? The PM2.5 in this year was recovered and higher than the other years. How large contribution of

the atmospheric circulation effect in your mind? Or the high PM2.5 is mainly sourced from the emission.

**Response 5:** The atmospheric circulation pattern in 2016 winter (Dec. 2016 to Feb. 2017) is almost the most unfavorable for the air pollutants diffusion based on our circulation classification, with the most frequent occurrence of unfavorable circulation types and second lowest frequency of favorable circulation types. The unfavorable circulation pattern in 2016 winter is partly responsible for its obvious rebound in PM2.5 concentration. In contrast, atmospheric condition in 2017 winter has the most frequent favorable and relative infrequent unfavorable circulation types, which is benefit for the significant decrease in PM2.5 concentration from 2016 to 2017. Except for 2016, the annual mean air pollutants concentrations have begun steadily reducing since 2013, which indicates the effects of emission reduction. Admittedly, it would go a long way toward dealing with the overall treatment of the air pollution, and the current occurrences of air pollution episodes are strongly depended on the meteorological background.

---

## Short Comment (SC2) · 27 Apr 2020

**Review 2**

**Comment:** This study makes a full investigation about the effects of atmospheric circulations on the interannual variation of PM2.5 over the Jing-Jin-Ji region, which is interesting and valuable to both science community and the society. It defines six types of atmospheric circulations and reveals their roles (favor or unfavor) to the PM2.5 concentration. In principle, the paper is a good contribution to the science community and worthy for publication after a minor revision as suggested below.

**Response:** Thank you for your positive comments and valuable suggestions.

**General Comments:**

**Comment 1:** It is always a puzzle regarding the relative contribution from emissions, meteorology, climate and topography to the aerosols observed. The authors found the results in Line 365-375, regarding which I have two questions here. The first one, also the most important one, the relative contribution from meteorological contribution found here is ~37% for most stations, which is much higher than the values found by other studies (~10%), then why? May you please give an explanation? The second one, with the same emission map, the decrease of PM2.5 from simulations between 2017 and 2016 is larger than that from observation, which seems to me that it implies more emissions in 2017 than in 2016. Is this true or possible?

**Response 1:** Most of the existing researches we involved in Introduction are focused on the evaluation of the relative contribution from emission and other elements during the whole period of Clean Air Action from 2013 to 2017 or 2018. The average contribution of meteorological conditions accounts for about 10% of the improvement of recent air quality. But the primary concern of our work is to investigate the effects of meteorological elements on the interannual variation of air quality, the magnitude of which may be larger than the multiyear averaged value. Based on the occurrence frequency of circulation types in the study period, the atmospheric circulation pattern in 2016 and 2017 winters are the most unfavorable and most favorable for the diffusion of air pollutants, respectively. Therefore, the two consecutive years of 2016 and 2017 are taken as the case of model simulation, and the magnitude of the contribution of meteorological conditions during the two years may be higher than the results of other studies.

The averaged observed PM2.5 difference between 2016 and 2017 winter is -37.7% at the 114 stations over Jing-Jin-Ji region. The model simulations are set with the same emission inventory driven by the meteorological fields of 2016 and 2017, respectively. The simulated PM2.5 difference between 2016 and 2017 can be attributed to the contribution of meteorological variation. The PM2.5 concentration difference from simulations between 2016 and 2017 is -22.6% at the 114 observation stations, which is lower than the magnitude of observed value. Therefore, the difference of meteorological fields between 2017 and 2016 could explain 60% of the 37.7% decrease in PM2.5 concentration, which suggests the emission reduced by 15% (40% of the 37.7% decrease) from 2016 to 2017.

**Comment 2:** Another thing is that the aerosol pollution is often coupled with the meteorology, causing non-linear relationships between aerosol pollution and meteorology or aerosol emissions. In other words, the relative contribution from both factors could vary with the air pollution cases. How could you account for this coupled effect?

**Response 2:** We agreed with the reviewer that there is feedback between aerosol and meteorology from the perspective of radiation and cloud. The fully coupled "online" WRF-Chem model has been used to evaluate the effects of meteorology on aerosols in this study, which includes the coupled physical and chemical processes such as transport, deposition, chemical transformation, photolysis and aerosol interaction with radiation and cloud. The chemistry module is turned on in the simulation, with RADM2 chemical mechanism and MADE/SORGAM aerosols. Some parameters related to direct and indirect effects of aerosol are also configured as follows, i.e., feedback from aerosol to radiation (aer_ra_feedback=1), feedback from the parameterized convection to the atmospheric radiation and photolysis (cu_rad_feedback=.true.), microphysics scheme (mp_physics=SBU-YLin), wet scavenging (wetscav_onoff=1) and cloud chemistry (cldchem_onoff=1). The effects of the no-linear feedback between aerosol pollution and meteorology have been simulated in the model, the results of which are combined into the meteorological factor contribution in this study. The specific selection of parameterization schemes is added in Lines 167-170 in this revised version.

In terms of the response of aerosol emission to the variation of aerosol, we used the online calculation of Gunther biogenic emissions parameterization scheme. However, some emission modules are turned off to reduce the computation cost, i.e., dust, sea salt, dimethysulfide, biomass burning and wildfires, which would result in the uncertainty of simulated PM2.5 mass concentrations. Some additional discussions about the

uncertainty of simulation are added in Lines 431-439 in the revised version.

**Minor comments:**

**Comment 1:** Line 17 and 43, full spell should be provided for PM2.5 when first used.

**Response 1:** Thanks for your reminder. We add the full spell of "PM2.5" in Lines 20-21.

**Comment 2:** Line 44-47, regarding the aerosol effect on climate by changing the surface radiation balance, four more references are recommended, Garrett and Zhao (2006, DOI: 10.1038/nature04636) and Zhao and Garrett (2015, doi:10.1002/2014GL062015) showed the aerosol's strong warming effect in the winter Arctic through increasing cloud thermal emissivity; Zhao et al. (2020, https://doi.org/10.1093/nsr/nwz184) showed the impacts of aerosols on the weather and climate by changing the radiation over the Tibetan Plateau; and Yang et al. (2018, DOI: 10.1016/j.atmosres.2018.04.029) showed the cooling effect of aerosols to Hongkong region climate during past 30 years.

**Response 2:** Thanks for your information. We involved these reference in the revised version.

**Comment 3:** Line 47, "pollutions" -> "Pollution".

**Response 3:** Revised as suggested.

**Comment 4:** Line 42, "air quality" -> "the air quality".

**Response 4:** Revised as suggested.

**Comment 5:** Line 51-52, you may change the second "strengthening" to "improving".

**Response 5:** Revised as suggested.

**Comment 6:** Line 64-67, One more reference could be also cited, which show significant improvement of air quality in five typical cities in China during recent several years, along with detailed discussions about the potential reasons for pollutions in these cities, Zhang et al. (2019, https://doi.org/10.1007/s13143-019-00125-w).

**Response 6:** Thanks for your information. The reference is added in Lines 69-70 in the revised version.

**Comment 7:** Line 69-72, these are true, which also include the dilution due to increasing the planetary boundary layer (Yang et al., 2016, doi:10.1002/2015JD024645), exchange of polluted and clean air, and hygroscopic growth of aerosols (Sun et al., 2019, DOI: 10.1029/2019EA000717; Zhao et al. 2018, https://doi.org/10.1007/s00376-017-7069-3). Moreover, Garrett et al. (2010, https://doi.org/10.1111/j.1600-0889.2010.00453.x) demonstrates the importance of long-range transport and wet scavenging to the aerosol amount in the Arctic; Sun et al. (2019) showed the relative roles of wet scavenging and hygroscopic growth the aerosols in Beijing, and Zhao et al. (2018) showed the fast growth of fine aerosols particles in Beijing.

**Response 7:** Thanks for your suggestion. We improve the description of meteorology effects on the evolution of air pollution in Lines 66-67 and Lines 75-77.

**Comment 8:** Line 72-76, regarding the climate signals, this is important. One thing I am not sure is how to differ this with short-term meteorological influence. The other is that Chen et al. (2019, https://doi.org/10.1007/s00382-019-04706-3) suggested that the Arctic warming have a strong tele-connection with mid-latitude air pollution (aerosol amount). For example, an increase in Arctic surface temperature in summer is associated with enhanced air pollution in Asia in winter.

**Response 8:** To distinguish the meteorological factors of different time-scale effects on the ambient air pollution, the general background atmospheric patterns are determined by the climate signals, which will stimulate the favorable diffusion circulation or not; but the actual occurrence frequency of favorable or unfavorable

circulations types are in fact of short-term meteorological conditions. In my opinion, the short-term meteorological elements control the evolution of most air pollution episodes, and climate signals influence the inter-annual and decadal anomaly of local air quality.

**Comment 9:** Line 80, "contribution to" -> "contribution from".

**Response 9:** Revised as suggested.

**Comment 10:** Line 106-108, since you are focusing on the winter time PM2.5 mass concentration over the BTH region, I would suggest to add a short paragraph to describe the wintertime PM2.5 pollution over the BTH region in the introduction part.

**Response 10:** Thanks for your suggestion. We add some description in Lines 111-116: *China's air quality shows obvious seasonal and regional distributions, with more frequent severe air pollution episodes in winter time and higher air pollutant concentrations in eastern China. As one of the three key regions in the Clean Air Action, lots of mitigation measurements have been taken over BTH region in recent years, which results in the significant improvement of local air quality, especially in winter time. But the relative contribution from meteorological factors are still unclear.*

**Comment 11:** Line 115, the region is also defined in Figure 2, why not using Figure 2?

**Response 11:** Thanks for your reminder. We revised it to Figure 2 in this version.

**Comment 12:** Line 116-117, Does this imply that the daily data is set as missing when the missing data is more than 40% in a day?

**Response 12:** Yes, the original PM2.5 data is hourly scale, which is averaged to daily mean with daily valid data more than 60%. We reorganized the sentence as "*Daily PM2.5 data is set as missing when the valid hourly data on the specific day is less than 40%.*"

**Comment 13:** Line 119-120, what do you mean "nonlinear methods" here?

**Response 13:** The nonlinear methods refer to some clustering methods based on neural network and deep learning, such as Cavazos (2000) investigate the extreme wintertime precipitation using the Self-Organizing Maps. We add this reference in the revised version.

**Comment 14:** Line 125, "Zhang et al. (Zhang et al., 2012)" -> "Zhang et al. (2012)"

**Response 14:** Revised as suggested.

**Comment 15:** Line 126-127, what do COST and PM mean here?

**Response 15:** COST is the abbreviation for European Cooperation in Science & Technology. We revise the PM to particulate pollution in this version.

**Comment 16:** Line 139, NCEP FNL should be fully spelled when first used.

**Response 16:** We add the full name of NECP FNL in the revised version.

**Comment 17:** Line 151-153, This is true. However, it just represents one case with different meteorology (2016 vs 2017). You may add one sentence to assume that this result is used to represent the typical value of meteorological contribution to PM2.5 concentration.

**Response 17:** Thanks for your suggestion. We revised the sentence as *Thus, the difference in the simulated PM2.5 concentrations between the 2016 and 2017 winters could be attributed to the meteorological variation, which can be assumed as a typical value of meteorological contribution to the interannual variation of PM2.5 concentrations.*

**Comment 18:** Line 157-158, Why do you use so long time to spin-up (15 days)? May you please briefly explain?

**Response 18:** We made a double check about the configuration of the model simulation, and found the start time is 23 November, which indicates one week to spin up. The simulated and observed meteorological fields from November 23$^{th}$ to the end of December in 2016 are shown here. Most of the simulated meteorological variables are consistent with observations after one week spin up run. We revised the description in the main text.

[Figure]

Figure r1. The observed and simulated air temperature (T2), sea level pressure (SLP), relative humidity (RH) and 10 m wind speed (wind speed) over Shijiazhuang, Beijing, Tianjin, Xuzhou and Shanghai during Nov. 23 to Dec. 31 in 2016.

**Comment 19:** Line 163, "Dominate" -> "Dominant".

**Response 19:** Revised as suggested.

**Comment 20:** Line 177-178, "the accumulate" -> "accumulate". In other word, remove "the".

**Response 20:** Revised as suggested.

**Comment 21:** Line 273, "in the last section": do you mean "this section"?

**Response 21:** We revised "in the last section" to "in section 3.1"

**Comment 22:** Line 299-301, delete either "although" or "but"

**Response 22:** We delete "but" in this version.

**Comment 23:** Line 311-313, "when the favorable circulation duration shorter . . ." -> "when the favorable circulation duration is shorter . . ."

**Response 23:** Revised as suggested.

**Comment 24:** Line 403-404, I would suggest "The 2020 is the key and target year for the three-year action to win the battle for a blue sky goal set in 2018".

**Response 24:** Revised as suggested.